

# A multi-objective path optimization method for plant protection robots based on improved A*-IWOA

Jing Niu[1,*], Chuanyan Shen[1,*], Lipeng Zhang[2], Qijun Li[1] and Haohao Ma[1,3]

[1] School of Mechatronics and Automotive Engineering, Tianshui Normal University, Tianshui, China
[2] School of Vehicle and Energy, Yanshan University, Qinhuangdao, China
[3] Department of Mechanical and Manufacturing Engineering, University Putra Malaysia, Serdang, Malaysia
* These authors contributed equally to this work.

Corresponding author
Jing Niu, sensily@163.com

## ABSTRACT

**Background:** The widespread adoption of plant protection robots has brought intelligent technology and agricultural machinery into deep integration. However, with advances in robotic autonomy, the energy that robots can carry remains limited due to constraints on battery capacity and weight. This limitation restricts the robots' ability to perform tasks continuously over extended periods.

**Methods:** To address the challenges of achieving low energy consumption and efficiency in path planning for plant protection robots operating in mountainous environments, a multi-objective path optimization approach was developed. This approach combines the improved A* algorithm with the Improved Whale Optimization Algorithm (A*-IWOA), utilizing a 2.5D elevation grid map. First, an energy consumption model was created to account for the robot's energy use on slopes, based on its kinematic and dynamic models. Then, an improved A* search method was established by expanding to an 8-domain diagonal distance search and introducing a cost function influenced by cross-product decision values. Using the robot's motion trajectory as a constraint, the IWOA algorithm was applied to optimize the vector cross-product factor (p) by dynamically adjusting population positions and inertia weights, to minimize both energy consumption and path curvature. Finally, in simulation and orchard scenarios, the application effects of the proposed algorithm were evaluated and compared against notable variants of the A* algorithm using the robot ROS 2 operating system.

**Results:** The experimental results show that the proposed algorithm substantially reduces the travel distance and enhances both path planning and computational efficiency. The improved approach meets the driving accuracy and energy consumption requirements for plant protection robots operating in mountainous environments.

**Discussion:** This algorithm offers significant advantages in terms of computational accuracy, convergence speed, and efficiency. Moreover, the resulting paths satisfy the stringent energy consumption and path planning requirements of robots in unstructured mountain terrain. This improved algorithm could also be replicated and applied to other fields, such as picking robots, factory inspection robots, and complex industrial environments, where robust and efficient path planning is required.

# INTRODUCTION

The integration of intelligent technology for agricultural machinery through the widespread use of agricultural robots has significantly enhanced production efficiency while reducing farmers' labor intensity. In recent years, technical issues in agricultural robotics have drawn increasing attention from scholars worldwide. Challenges remain in areas like visual navigation and trajectory decision-making. Advanced visual navigation methods based on deep learning, such as convolutional neural networks (CNN) and support vector machines, have enabled agricultural robots to more accurately and efficiently recognize and operate on unstructured terrain, including farmland and orchards (*Mao et al., 2020*). However, effective trajectory planning requires a sophisticated understanding of complex agricultural environments, and current spatial information processing methods based on visual navigation have certain limitations.

Path planning for agricultural robots, particularly when considering obstacle distribution, has become a focal point of research in smart agriculture. *Akyol & Alatas (2020)* categorized this research into classical algorithms like A*, intelligent optimization algorithms, and artificial intelligence-based approaches. While traditional algorithms such as A* and Rapid Random Tree (RRT) have been refined over the years, optimizing paths in complex scenarios remains challenging. In their study, *Lin et al. (2023)* addressed high memory consumption and lengthy operation times in the A* algorithm by enhancing its heuristic function. *Zhang et al. (2022)* improved RRT path planning by incorporating Gaussian functions, improving the robot's maneuverability in tight corners. Intelligent optimization algorithms include ant colony optimization (ACO), genetic algorithm (GA), sparrow search algorithm (SSA), and whale optimization algorithm (WOA). For instance, *Meng et al. (2023)* integrated multi-objective optimization indicators such as path length, safety, and energy consumption into an enhanced ACO, achieving global path optimization. Additionally, artificial intelligence approaches, including neural network algorithms, machine learning, and deep learning algorithms, represent major advancements for path planning in both static and dynamic environments.

As robot autonomy advances, limited battery capacity and weight constraints restrict the energy available to robots (*Miao et al., 2021*), posing a challenge to their ability to perform prolonged tasks. Reducing energy consumption has thus become critical to enhancing the operational efficiency of robots, and a large block of studies has investigated this aspect from different perspectives. *Jones & Hollinger (2017)* proposed an energy-efficient path planning algorithm that incorporates a stochastic optimization approach to estimate energy consumption in uncertain and disturbed environments. Additionally, *Zhang, Zhang & Zhang (2020)* proposed a two-layer approach to optimal energy consumption path planning. First, multiple paths are generated using traditional planning methods, such as RRT. Then, energy consumption for each path is estimated by employing

an energy consumption model, allowing for the selection of the optimal energy consumption path. The main difference between these methods lies in their underlying energy consumption models.

In mountainous orchard environments, factors such as spacing between fruit trees, terrain undulations, leaves, weeds, seasons, and lighting can significantly impact the accuracy of machine vision (*Miao & Li, 2010*) running time and trajectory. This, in turn, affects the robots' running time and trajectory. Addressing the impact of terrain variation on visual navigation accuracy is essential in trajectory planning for agricultural robots.

Plant protection robots enable efficient obstacle avoidance path planning by sensing, detecting, and identifying obstacles using onboard sensors. As typical unstructured environments, farmland and orchards present challenges such as uneven terrain and complex distribution of obstacles for complicate path planning. *SoundraPandian & Mathur (2010)* moved path points further from obstacles by using mixed A* for safe distance path planning. In their investigation, *Zhen et al. (2007)* selected dynamic points along the line between the robot and the target point as feature vectors, utilizing the COA (Coyote Optimization) algorithm iteratively until the target point was reached. *Yuan et al. (2020)* developed a combination matrix incorporating energy consumption and motion distance models, applying it to the Dijkstra algorithm for path planning (*Yuan et al., 2020*; *Wang, 2012*; *Meng et al., 2023*). An inspired algorithm combining Fokker-Planck equation and the intermittent diffusion process was also introduced to incorporate energy consumption constraints, optimizing energy-efficient path planning in resource-limited situations (*Zhai, Egerstedt & Zhou, 2022*). Moreover, *Zakharov, Saveliev & Sivchenko (2020)* proposed the Local Roughness Local Height Difference A* (LRLHD-A*) algorithm for optimal energy-efficient path planning of robots in three-dimensional map environments. However, traditional energy consumption models, given the dynamic terrain and robot movement interactions, often lack accuracy. *Lambert et al. (2024)* leveraged deep meta-learning algorithms, training only limited terrain perception data, to generate more accurate energy consumption models.

The A* algorithm is a classic method for finding optimal paths in static environments, making it well-suited for orchard settings dominated by static obstacles. Its core function is to evaluate the search cost of each state node, with the search cost primarily depending on the length of the selected trajectory. As is well known, considering terrain undulations in search costs provides a more accurate assessment than assuming a flat, horizontal surface. Therefore, it is necessary to optimize search costs for terrain undulations in A*. The WOA algorithm has demonstrated good timeliness and robustness in multi-objective optimization problems and can meet the computational efficiency requirements of the real-time operation of plant protection robots. Thus, WOA can be introduced into A* path planning to address multi-objective optimization problems such as trajectory length, trajectory smoothness, and running time.

Currently, most trajectory-decision algorithms implemented in robotics use 2D or 3D grid maps. Positioning and navigation methods based on 2D grid maps are well-studied and widely adopted across various scenarios. Meanwhile, 3D grid maps can better represent outdoor, uneven, or rough terrain. However, creating accurate 3D surface

models in such environments is challenging and computationally expensive, making it a relatively exploratory approach in robotics. In this study, based on a 2.5D elevation grid map, a work energy consumption model that accounts for additional energy consumed by robots on slopes is established. A path planning algorithm combining the improved A* algorithm with the Improved Whale Optimization Algorithm (A*-IWOA) is designed with kinematic constraints on the robot's motion trajectory as the boundary condition, ensuring a balance between energy consumption and trajectory smoothness in the robot's operational performance.

## MATERIALS AND METHODS

### Kinematic and energy consumption models of the plant protection robot

This research subject is a front-wheel differential-driven Ackermann steering plant protection robot operating in unstructured orchard environments. This robot's coordinate system (XYZ) is defined within the geodetic coordinate system ($X_0Y_0Z_0$), where the X-axis points directly in front of the robot, the Y-axis points leftward, and the Z-axis is perpendicular to the robot's moving platform, as shown in Fig. 1. Considering the effects of the robot's operational status and orchard terrain, this study establishes kinematic state-space equations for movements along the X-axis, Y-axis, and Z-axis, as well as lateral motion around the Z-axis.

### Kinematic model

This robot's motion state variable is defined as $X = [v_x, \omega]$, where $v_x$ is the velocity component along the X-axis and $\omega$ is the lateral angular velocity around the Z-axis. $U = [\omega_l, \omega_r]$ is defined as control variables, where $\omega_l$ and $\omega_r$ represent the angular velocities of the left and right drive wheels of the robot, respectively. The robot's motion state equation can be expressed as:

$$\begin{bmatrix} v_x \\ \omega \end{bmatrix} = \frac{r}{B} \begin{bmatrix} \frac{B}{2} & \frac{B}{2} \\ -1 & 1 \end{bmatrix} \begin{bmatrix} \omega_l \\ \omega_r \end{bmatrix} \tag{1}$$

$$z = \sigma \tag{2}$$

where r is the wheel radius of the robot, B is the track width, z is the Z-axis displacement of the robot, and $\sigma$ is the road elevation.

The kinematic model of the robot, represented by the unilateral driving wheel motion state, is defined as *Zhang et al. (2023)*:

$$\mathbf{M}\ddot{\boldsymbol{\theta}} + \mathbf{C}(\boldsymbol{\theta}, \dot{\boldsymbol{\theta}}) + \mathbf{G}(\boldsymbol{\theta}) = \mathbf{T} \tag{3}$$

where $\boldsymbol{\theta} = [\theta_l, \theta_r]$, $\dot{\boldsymbol{\theta}} = [\omega_l, \omega_r]$, and $\mathbf{T} = [T_l, T_r]$.

Here, M represents the inertia matrix of the robot's driving wheels, $\mathbf{C}(\boldsymbol{\theta}, \dot{\boldsymbol{\theta}})$ is the ground rolling resistance moment matrix of the driving wheel, and $\mathbf{G}(\boldsymbol{\theta})$ is the gravity matrix of the robot. Also, $\boldsymbol{\theta}$ denotes the angular displacement vector for the left and right driving wheels, $\dot{\boldsymbol{\theta}}$ is the angular velocity vector for the left and right driving wheels, and $\ddot{\boldsymbol{\theta}}$ is

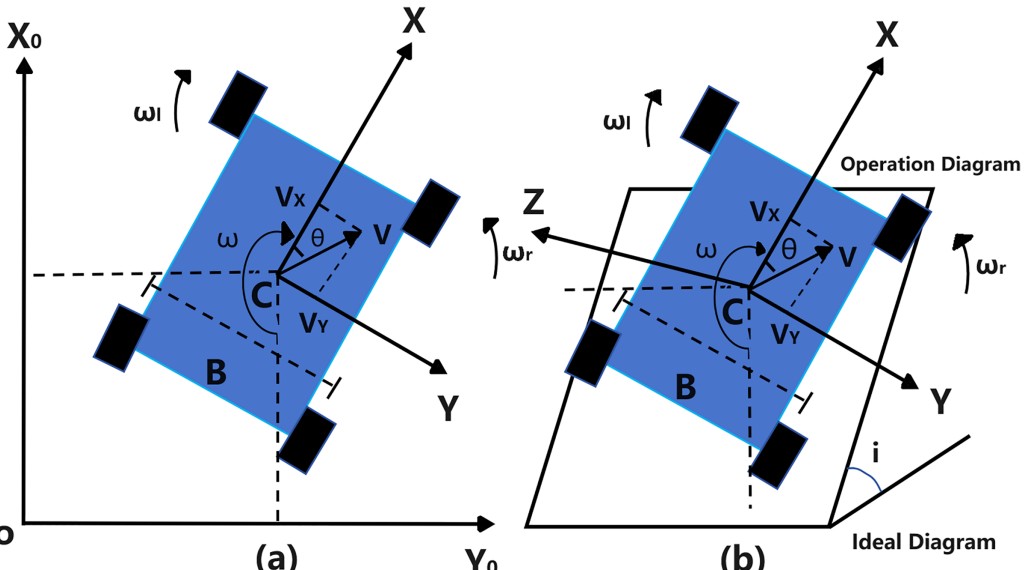

**Figure 1 Front wheel differential drive Ackermann steering robot coordinate system.** (A) 2D schematic operation diagram; (B) 3D space schematic operation diagram.

the angular acceleration vector for the left and right driving wheels of the robot. **T** represents the output torque matrix for the left and right driving wheel motors.

The relationship between the robot's control variable U and the output torque vector T of the driving wheels can be derived from Eqs. (1) to (3), laying the foundation for developing its energy consumption model.

## Energy consumption model

The employed fully electric-driven plant protection robot uses power batteries as the power source, with the motor controller managing motor speed through pulse-width modulation (PWM) (modulation and demodulation) methods. Part of the power output of the driving motor is consumed by the internal resistance of the battery, while the remaining portion powers the robot's moving platform.

Given the relatively low movement speed of the plant protection robot and the limited tire contact area, air resistance and rolling resistance are negligible. However, the orchard's uneven terrain and the robot's considerable mass mean that ramp resistance cannot be ignored (*Yin et al., 2019*). Therefore, this article defines energy consumption as the sum of battery internal resistance loss and robot ramp resistance loss.

With a constant robot load and motor output torque, and based on the direct proportionality between motor torque and armature current, $I_l$ and $I_r$ are considered constant values. The armature voltages for the left and right drive motors are expressed as:

$$U_l = I_l R_B + K_M i_0 \omega_l \tag{4}$$
$$U_r = I_r R_B + K_M i_0 \omega_r \tag{5}$$

where $I_l$ and $I_r$ are the armature currents of the left and right drive motors, respectively, $R_B$

is the internal resistance of the power batteries, and $U_l$ and $U_r$ denote the armature voltages for the left and right drive motors, respectively. $K_M$ represents the back electromotive force coefficient of the driving motor, and $i_0$ is the transmission ratio of the motor reducer.

The power outputs of the left and right drive motors are expressed by Eqs. (6) and (7), respectively.

$$P_l = U_l I_l \tag{6}$$
$$P_r = U_r I_r \tag{7}$$

Substituting Eqs. (4) and (5) into Eqs. (6) and (7), the output power expressions become:

$$P_l = I_l^2 R_B + K_M i_0 \omega_l I_l \tag{8}$$
$$P_r = I_r^2 R_B + K_M i_0 \omega_r I_r \tag{9}$$

The internal resistance loss of the battery can thus be expressed as follows:

$$Q_B = \left( I_l^2 + I_r^2 \right) R_B \sum_{i=1}^{N} \frac{ds_i}{v_x} \tag{10}$$

where N is the number of state nodes in the path search node space, and $ds_i$ is the distance between adjacent state nodes.

Assuming the longitudinal ramp angle of the road surface is $\alpha$, the ramp resistance loss can be expressed by the following equations:

$$F_i = mg\tan\alpha \tag{11}$$
$$\tan\alpha = D \cdot \frac{z_{i+1} - z_i}{d_0}, \quad z_{i+1} \geq z_i \tag{12}$$

where $m$ is the total mass of the robot, ignoring mass changes during operation. The coefficient $D$ in Eq. (12) depends on the search logic, where $D = 1$ is used for straight line search and $D = 0$ for diagonal search. $d_0$ denotes the distance between the centers of adjacent cells in a grid map.

Thus, the ramp resistance loss can be further expressed as:

$$Q_i = \frac{mgD}{d_0} \sum_{i=1}^{N} (z_{i+1} - z_i) \frac{ds_i}{v_x} \tag{13}$$

By summing Eqs. (10) and (13), the robot's energy consumption model is obtained as:

$$Q = \sum_{i=1}^{N} [A + B(z_{i+1} - z_i)] \frac{ds_i}{v_x} \tag{14}$$

where A and B are constant coefficients related to the robot's structural parameters and node search logic, respectively.

## An improved A$^*$ path searching method based on the constraints of operation conditions

The A* algorithm is a heuristic search method used to find the optimal path in environments with static obstacles, searching within the robot's motion state space (*Min et al., 2021*). First, it evaluates the cost at each search position to identify the state node with the lowest cost. Then, it traverses the entire state space until finding the optimal solution, at which point it terminates the search cycle.

In this heuristic search process, evaluating the cost state nodes is crucial, typically expressed by the following cost function:

$$f(n) = g(n) + h(n) \tag{15}$$

where $f(n)$ is the cost function from the initial state through state $n$ to the target state, $g(n)$ is the actual cost from the initial state to state $n$ in the state space, and $h(n)$ is the estimated cost of the optimal path from state $n$ to the target state.

In 2D grid maps, three common $h(n)$ functions are Euclidean distance, Manhattan distance, and diagonal distance, as illustrated in Fig. 2 (*Mittelmann & Peng, 2010*). While Euclidean distance is the shortest, it may reduce search efficiency in complex environmental maps. Manhattan distance offers simple logic but results in a longer path. In contrast, the diagonal distance method generally performs best in terms of both search path distance and efficiency (*Shi et al., 2023*), making it a preferred method for optimal path planning.

## Enhancement of A$^*$ based on the cost function of vector cross-product winning value

This study addresses the limitations of 2D grid information in describing the working environment and the high demand for computing resources of 3D-occupied grid maps using an octree structure (*Wu et al., 2022*). Additionally, considering that energy consumption in mountainous areas significantly impacts path selection, this article uses a 2.5D elevation grid map to better capture the robot's working environment with higher accuracy. This approach adds height information of the grid center point in a 2D map with only horizontal and vertical coordinates, as shown in Fig. 3. Figure 3A displays the terrain conditions of the robot's passage area with 3D grid node coordinates. Also, Fig. 3B uses hues of different grids to represent the vertical height from the horizontal plane at the grid's center point, denoted by $z_n$. Hue differences indicate variations in the vertical coordinates across grid nodes, providing efficient environmental representation with lower maintenance costs and higher real-time performance. In the simulation results, the planned path obtained in a 2.5D elevation grid map differs markedly from the 2D grid environment, where the vertical height of the mountains is disregarded.

According to the 8-domain diagonal distance search method (*Saadatzadeh, Ali Abbaspour & Chehreghan, 2023*), a cost function with a cross-product winning value is introduced to bias the planned path toward a straight line from the initial point to the target point (*Bays et al., 2024*), as illustrated in Fig. 4. The specific definitions are as follows:

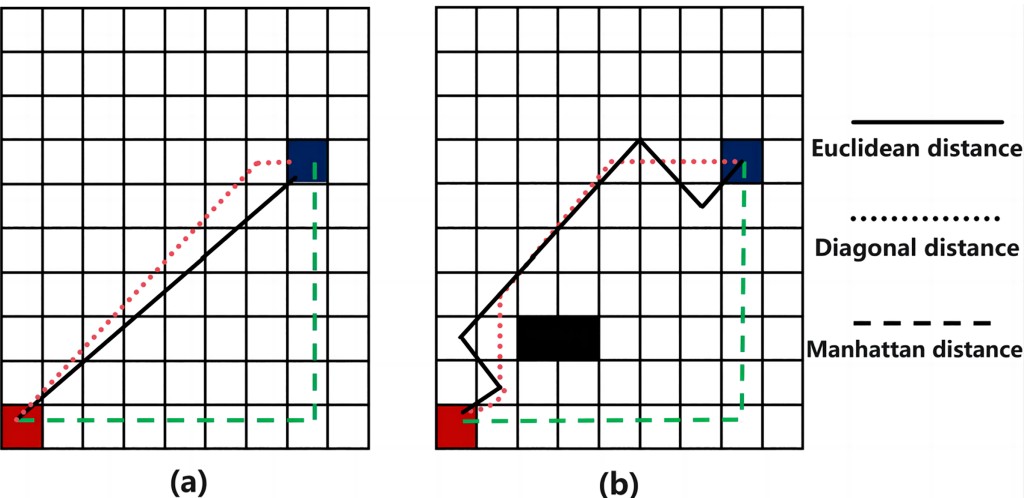

**Figure 2 Path planning results comparison of three distance functions.** (A) Non-obstacle situation; (B) obstacle situation.

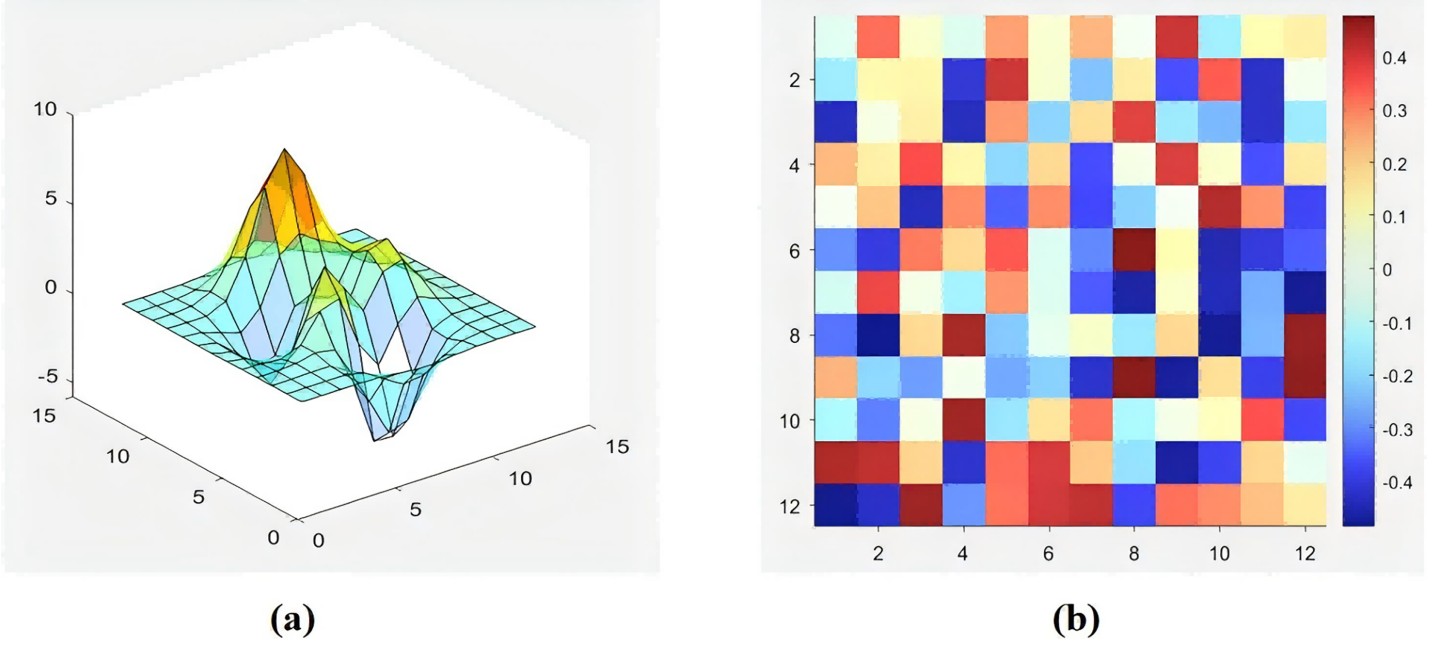

**(a)**   **(b)**

**Figure 3 2.5D elevation grid map.** (A) Vertical height of mountain orchard ground; (B) 2D grid map.

$$dx1 = x_n - x_{goal} \tag{16}$$

$$dy1 = y_n - y_{goal} \tag{17}$$

$$dz1 = z_n - z_{goal} \tag{18}$$

$$dx2 = x_{start} - x_{goal} \tag{19}$$

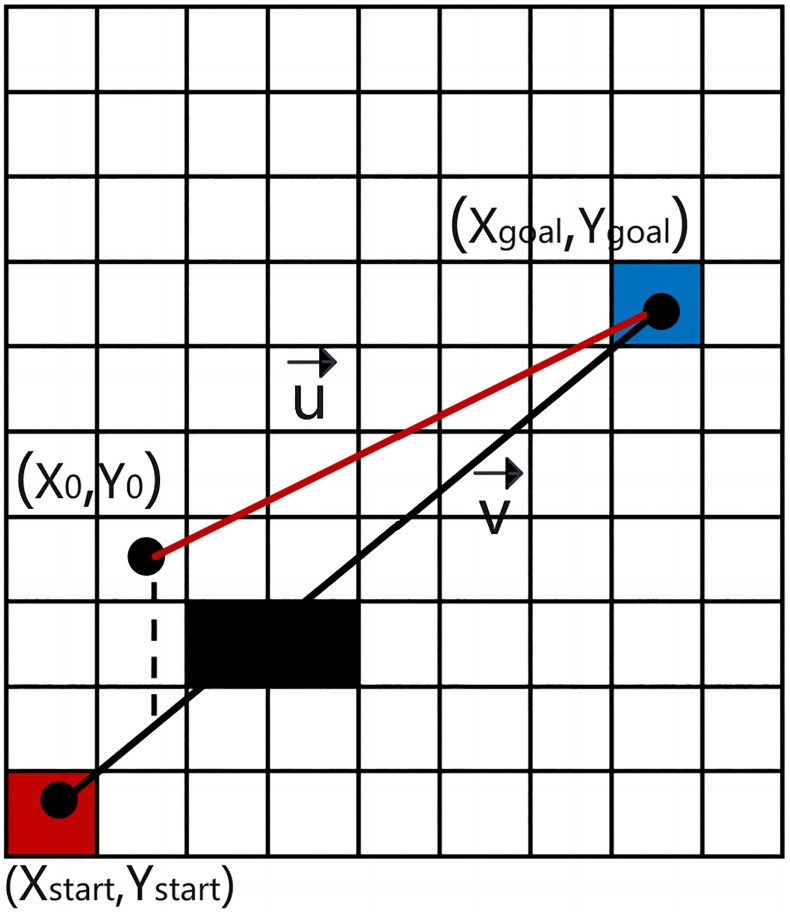

**Figure 4 Definition of the cross product.**

$$dy2 = y_{start} - y_{goal} \tag{20}$$
$$dz2 = z_{start} - z_{goal} \tag{21}$$

In this context, **u** and **v** represent the vector from the current point to the target point and the vector from the starting point to the target point, respectively, as represented by Eqs. (22) and (23).

$$\mathbf{u} = (dx1, dy1, dz1) \tag{22}$$
$$\mathbf{v} = (dx2, dy2, dz2). \tag{23}$$

To measure the deviation of the planned straight path between the current node and the starting and target points, the cross-product vector of **u** and **v** can be defined as follows:

$$\mathbf{u} \times \mathbf{v} = \begin{bmatrix} i & j & k \\ dx1 & dy1 & dz1 \\ dx2 & dy2 & dz2 \end{bmatrix} = (dy1*dz2 - dy2*dz1, dz1*dx2 - dx1*dz2, dx1*dy2 - dy1*dx2) \tag{24}$$

On this basis, the vector cross-product winning value is defined by Eq. (25).

$$cross = \sqrt[2]{(dy1 * dz2 - dy2 * dz1)^2 + (dz1 * dx2 - dx1 * dz2)^2 + (dx1 * dy2 - dy1 * dx2)^2}. \quad (25)$$

The vector cross-product winning value evaluates the positional difference between two vector spaces (*Wang, Wang & Liu, 2024*). In this regard, the greater the overlap between the two vector spaces, the smaller this value, and conversely, the larger the value.

Thus, by incorporating the vector cross-product winning value, the cost function is redefined as follows:

$$h(n) = 1 + cross * p \quad (26)$$

where $(x_n, y_n, z_n)$, $(x_{start}, y_{start}, z_{start})$, and $(x_{goal}, y_{goal}, z_{goal})$ are the coordinates of the current state node, the starting point, and the target point, respectively; $p$ is the vector cross-product weight factor.

In Fig. 4, the parallelogram area formed by **u** and **v** vectors represent the cross value. It is noted here that the greater the deviation between the current path and the straight path from the start to the target, the larger the cross value becomes. According to the cost function's tendency, path nodes are chosen in directions closer to the straight path. When the value of $p$ is properly chosen and no obstacles are present, A* can search through fewer state regions while finding efficient paths. However, if a fixed $p$-value is used in the presence of many obstacles, A* may yield irregular results, as shown in Fig. 5. Therefore, in "Performance testing of the improved IWOA algorithm" below, the intelligent optimization algorithm WOA is employed to optimize the vector cross-product factor $p$ to minimize robot operation energy consumption and path curvature.

### Constraints of operation trajectory

As shown in Fig. 6, the operation trajectory of the plant protection robot is divided into straight and curved segments. The search logic for the straight section is straightforward easily. Points $Q_0 \sim Q_6$ in the figure represent seven consecutive state nodes within a specific curved trajectory segment, where $Q_0 (x_0, y_0)$ and $Q_6 (x_6, y_6)$ are the start and target points, respectively. It is noted here that $Q_1 (x_1, y_1)$ and $Q_5 (x_5, y_5)$ serve as segmentation points. To simplify the turning logic, the trajectory is symmetrically distributed along the center line. Adjusting the positions of $Q_2 (x_2, y_2)$ and $Q_4 (x_4, y_4)$ can enhance the smoothness of the trajectory. Due to the robot's unique working environment and structural constraints, the following requirements are established for the motion trajectory in the path planning process:

(1) The curvature of any point on the trajectory is constrained to $\rho \leq \dfrac{1}{R_{min}}$, where $R_{min}$ is the minimum turning radius of the robot.

(2) The front wheel turning angle of the robot is constrained to $\delta \leq \delta_{max}$, where $\delta_{max} = \arctan \dfrac{L}{R_{min}}$.

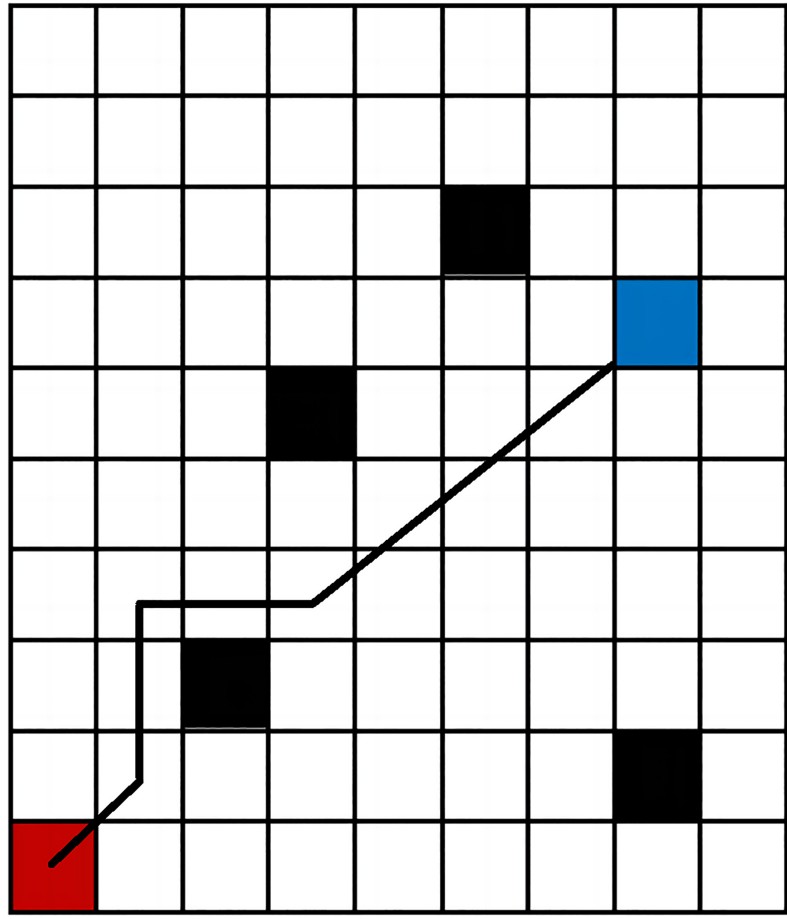

**Figure 5** The path of fixed p-factor cross product. 

(3) The trajectory curvature must be continuous. To avoid situations such as sharp turns and sudden stops, it is essential to maintain a continuous curvature of the trajectory. Thus, point $Q_2$ should be positioned above the line connecting $Q_1$ and $Q_3$; otherwise, excessive curvature change would hinder tracking (*Zhai et al., 2024*). Ignoring the effects of minor elevation parameters in a 2.5D elevation grid map, the curvature continuity condition can be expressed as:

$$y_2 \geq \frac{y_3 - y_1}{x_3 - x_1}(x_2 - x_1) + y_1 \tag{27}$$

(4) The angular velocity constraints of the robot front wheel can be given by:

$$\frac{d\delta}{dt} = \frac{d\left(arctan\dfrac{2Lsin\varphi}{l_d}\right)}{dt} \leq \omega_{max} \tag{28}$$

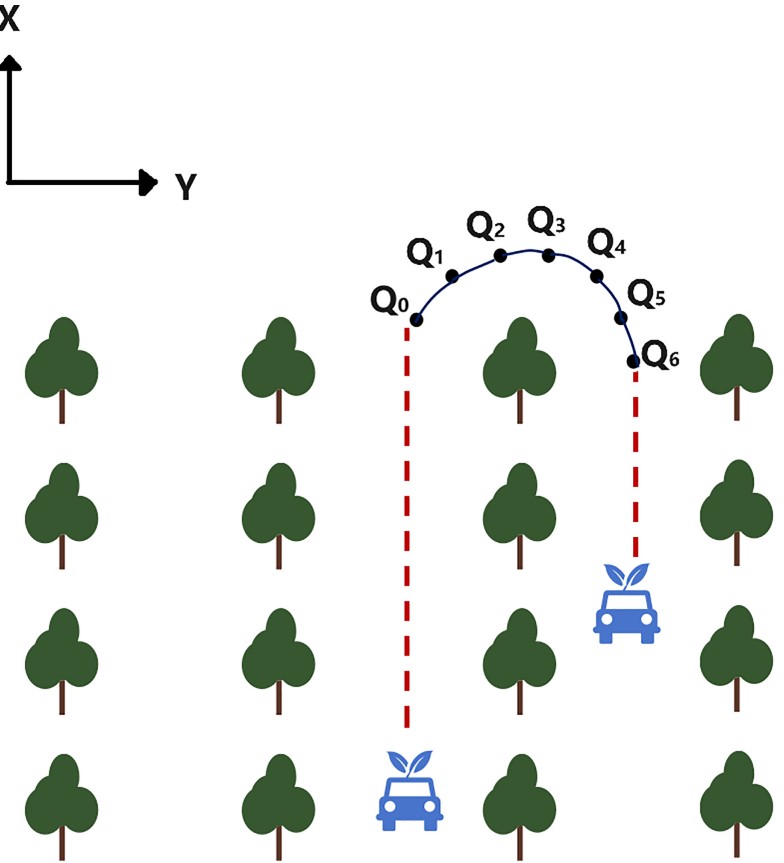

**Figure 6  Schematic diagram of the operation trajectory.**

This study utilizes cubic B-spline curves to fit trajectories (*Ardestani, Safdari & Mallah, 2023*). The above trajectory constraints can be summarized as follows:

$$
\begin{cases}
\rho(t) = \dfrac{y'(\boldsymbol{u})x''(\boldsymbol{u}) - x'(\boldsymbol{u})y''(\boldsymbol{u})}{(x'^2(\boldsymbol{u}) + y'^2(\boldsymbol{u}) + z'^2(\boldsymbol{u}))^{\frac{3}{2}}} \leq \dfrac{1}{R_{min}} \\[4mm]
y_2 \geq \dfrac{y_3 - y_1}{x_3 - x_1}(x_2 - x_1) + y_1 \\[4mm]
\dfrac{d\left(arctan\,\dfrac{2L sin\varphi}{l_d}\right)}{dt} \leq \omega_{max}
\end{cases}
\tag{29}
$$

where $\boldsymbol{u}$ is the node vector of the cubic B-spline curve, $l_d$ is the robot's forward viewing distance, and $\varphi$ is the heading angle between the robot's current position and the target point.

## Multi-objective optimization of vector cross-product weight factors

### Introduction to WOA

The WOA is a meta-heuristic algorithm that mimics the hunting behavior of humpback whales in the ocean. It simulates three stages of whale hunting: searching for prey,

surrounding targets, and spiral bubble net predation. Compared to other intelligent algorithms (*Huang et al., 2023*), it offers advantages like fewer parameters, simpler principles, and stronger multi-objective optimization capabilities. These three stages can be modeled mathematically as follows (*Rahimnejad et al., 2023*):

(1) Surrounding prey stage: Other individuals in the whale population update their positions and move closer to the optimal whale individual using Eqs. (30) to (34):

$$X_i^{t+1} = X_{best}^t - A \cdot D_1 \tag{30}$$

$$D_1 = \left| C \cdot X_{best}^t - X_i^t \right| \tag{31}$$

$$A = 2a \cdot r - a \tag{32}$$

$$C = 2 \cdot r \tag{33}$$

$$a = 2 - 2\frac{t}{T} \tag{34}$$

where $X_{best}^t$ is the position of the whale individual that has found the optimal solution in the t-th generation, $X_i^t$ is the position of the i-th whale in the t-th iteration, $D_1$ indicates the enclosing step size, A and C are coefficient vectors, $T$ is the maximum number of iterations, and $r$ is a random number between [0, 1].

(2) Bubble net attack stage: This stage simulates the process of whales forming a bubble net along a spiral to approach their prey, updating individual positions using Eq. (35), as follows:

$$X_i^{t+1} = D_2 \cdot e^{bl} \cdot cos(2\pi l) + X_{best}^t \tag{35}$$

where $D_2 = \left| X_{best}^t - X_i^t \right|$ represents the distance between the whale and its prey, $b$ is the spiral shape coefficient, and $l$ is a random number in the range of [0, 1].

(3) Searching for prey stage: WOA selects a random individual from the population as a target for position updates, updating the model as shown in Eqs. (36) and (37).

$$X_i^{t+1} = \begin{cases} X_{rand}^t - A \cdot D_3, & p < 0.5; \\ D_2 \cdot e^{bl} \cdot cos(2\pi l) + X_{best}^t, & p \geq 0.5 \end{cases} \tag{36}$$

$$D_3 = \left| C \cdot X_{rand}^t - X_i^t \right| \tag{37}$$

where $X_{rand}^t$ is a randomly selected individual position from the whale population.

### Improvement of IWOA based on dynamic adjustment of uniformly distributed population position and inertia weights

When initializing the WOA algorithm population, random generation of population positions can lead to uneven distribution, a limited search range, slow convergence speed, and susceptibility to local optima (*Yang & Liu, 2022*). To address these limitations, this article employs Circle mapping to generate uniformly distributed population positions, increasing the diversity of whale positions and enhancing WOA's performance (*Zhou et al., 2017*).

Circle mapping can be defined as follows:

$$X_i^{t+1} = mod\left( X_i^t + 0.2 - \frac{0.5}{2\pi} \sin\left(2\pi X_i^t\right), 1 \right) \tag{38}$$

where $X_i^t$ represents the position vector of the i-th whale in the population at the t-th position update.

Dynamic adjustment of the inertia weight of fitness $\omega$ is based on the $\Gamma$ inverse incomplete function (*Ogasawara, 2024*), with the specific form given by:

$$\omega = \frac{\omega_{max} - \omega_{min}}{\lambda} \times gammaincinc\left( \lambda, 1 - \frac{t}{T} \right) \tag{39}$$

where $\omega_{max} = 0.8$, $\omega_{min} = 0.3$, gammaincinc$(\lambda, a)$ is a MATLAB $\Gamma$ function defined as $\gamma(\lambda, a) = \int_0^\lambda e^{-t} t^{a-1} dt$, $\lambda (\lambda \geq 0)$ is a random variable, set to 0.2, t is the current iteration count, and T is the maximum number of iterations. Upon dynamic adjustment, the inertia weight $\omega$ decreases non-linearly as iteration progresses. On this basis, the improved IWOA position update formula is as follows:

$$X_i^{t+1} = \begin{cases} \omega \cdot X_{best}^t - A \cdot D, & |A| < 1, p < 0.5; \\ \omega \cdot X_{rand}^t - A \cdot D_{rand}, & |A| \geq 1, \ p < 0.5 \\ D \cdot e^{bl} \cdot cos(2\pi l) + \omega \cdot X_{best}^t, & p \geq 0.5 \end{cases} \tag{40}$$

where $X_{rand}^t$ represents a randomly selected position vector from the whale population at the t-th position update, $X_{best}^t$ is the optimal whale position vector from the whale population at the t-th position update, and p denotes the probability of choosing to reduce enclosure or update the spiral rotation position during whale hunting. Also, $D = \left| C \cdot X_{best}^t - X_i^t \right|$, $D_{rand} = \left| C \cdot X_{rand}^{*t} - X_i^t \right|$, where $X_{rand}^{*t}$ is a randomly selected whale position from the population. b denotes the constant of the spiral equation, set to 1 in this article. l is a random number in the range of $[-1, 1]$. A and C are two random parameters, defined as follows:

$$A = 2ar_1 - a, \quad C = 2r_2 \tag{41}$$

where $r_1$ and $r_2$ are random numbers in the range of $[0, 1]$, and a is a parameter that decreases from 2 to 0 as iterations increase, defined as:

$$a = 2 - 2t/T \tag{42}$$

The search steps follow the pseudo-code for the improved IWOA algorithm shown in Table 1.

### Process of multi-objective IWOA based on optimal solution evaluation

The distance between any two adjacent state nodes in the robot path is given by:

$$S_i = \sqrt{(x_{i+1} - x_i)^2 + (y_{i+1} - y_i)^2 + (z_{i+1} - z_i)^2} \tag{43}$$

During the operation of the plant protection robot, longitudinal speed changes are relatively minor and can be assumed to hold a constant value (*Rahimnejad et al., 2023*). Therefore, substituting Eq. (43) with Eq. (14) yields the energy consumption of the robot from the starting point to any other point:

**Table 1 Improved IWOA algorithm pseudo code.**

01  Set population size as N, maximum number of iterations as T

02  According to Formula (38), initialize the population position following the Circle map and calculate the fitness of each individual to determine the optimal individual position

03  Calculate the inertia factor according to Formula (39), and update A and C according to Formulas (41) to (42)

04  While (t < T)

05  for each individual

06  Calculation parameters $a$, $A$, $C$, $l$, p

07  if $p < 0.5$

08  if $|A| < 1$

09  Update individual position using Formula (40-1)

10  else

11  Update individual position using Formula (40-2)

12  end if

13  else

14  Update individual position using Formula (40-3)

15  end if

16  end for

17  Recalculate individual fitness according to Formula (29) boundary constraint processing

18  Update Best Individual

19  t = t + 1

20  end while

21  Output global optimal solution and optimal fitness

22  end

$$Q = \sum_{i=1}^{N} [A + B(z_{i+1} - z_i)] \frac{ds_i}{v_x} \tag{44}$$

It is noted here that when $z_{i+1} \geq z_i$ , $B \neq 0$; otherwise if $z_{i+1} < z_i$, $B = 0$.

The general parameter equation of the trajectory curve in three-dimensional space can be expressed as:

$$x = x(t), \ y = y(t), \ z = z(t) \tag{45}$$

The curvature at any point on the robot's operation path is expressed as:

$$\rho(t) = \frac{y'(t)x''(t) - x'(t)y''(t)}{(x'^2(t) + y'^2(t) + z'^2(t))^{\frac{3}{2}}}. \tag{46}$$

Two fitness functions are therefore established for the IWOA algorithm, as follows:

$$f_1(x_i, y_i, z_i) = Q, \quad f_2(x_i, y_i, z_i) = \rho(t). \tag{47}$$

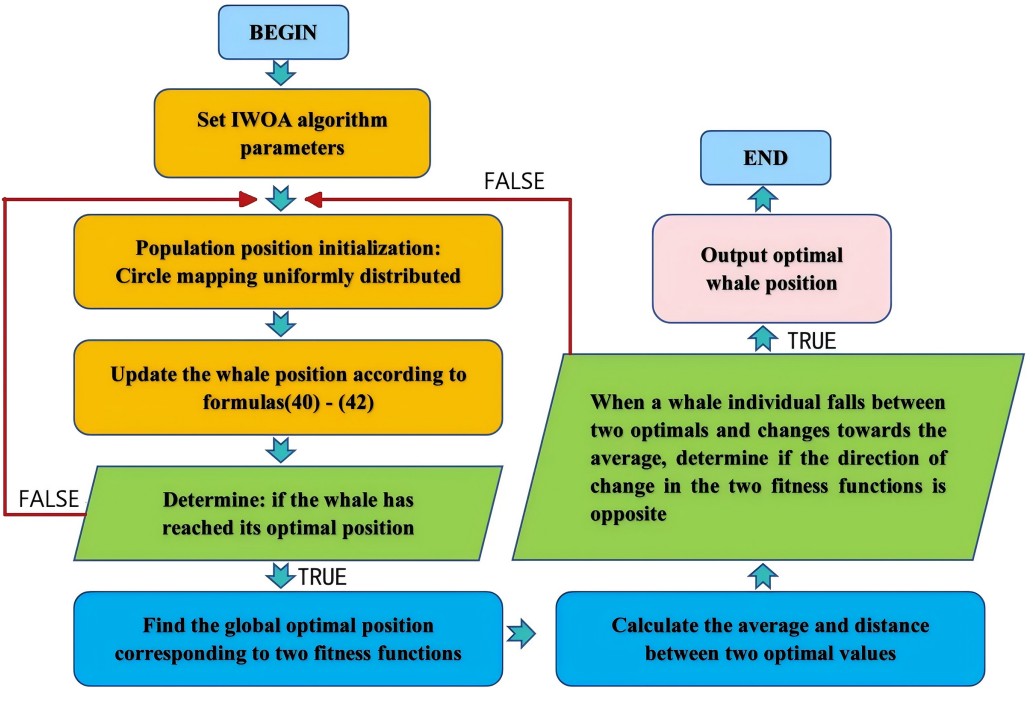

**Figure 7 IWOA multi-objective optimization process based on optimal selection.**

This study uses optimal solution evaluation logic to search for non inferior optimal solutions. In this optimal solution $f_1(x_i, y_i, z_i)$ and $f_2(x_i, y_i, z_i)$ change in different directions in updating. Ultimately, the whale's position is dispersed in a set of non inferior optimal solutions, which can prevent individual fall into the optimal solution region of a certain fitness function, reflecting the constraint relationship between the two fitness functions. The specific process is shown in Fig. 7.

# RESULTS

## Experiments and analysis

The experimental subject considered in this study is a wheeled plant protection robot independently developed at the author's university, shown in Fig. 8. As a visual sensor, the robot platform uses an OBI Zhongguang global shutter binocular depth camera with a depth frame rate of 90 fps. The 16-line LiDAR, Raytheon M10P, has a measurement radius of 30 m and a sampling frequency of 20,000 Hz. The processor is NVIDIA's Orin Nano NX 8 GB, operating on Ubuntu 18.04 LTS, with the overall functional design based on ROS 2. The robot operates at speeds of 2–5 km/h.

To verify the robustness of the proposed algorithm in this study and the improvements in path planning for typical job scenarios, the experiment consists of three parts. First, the improvement effect of the IWOA, which dynamically adjusts a uniformly distributed population position and inertia weight in a simulation environment, was compared to several variations of the WOA: the Lévy flight based on the Whale Optimization Algorithm ((LWOA)) to optimize and update position (*Zhao & Peng, 2023*), the Whale

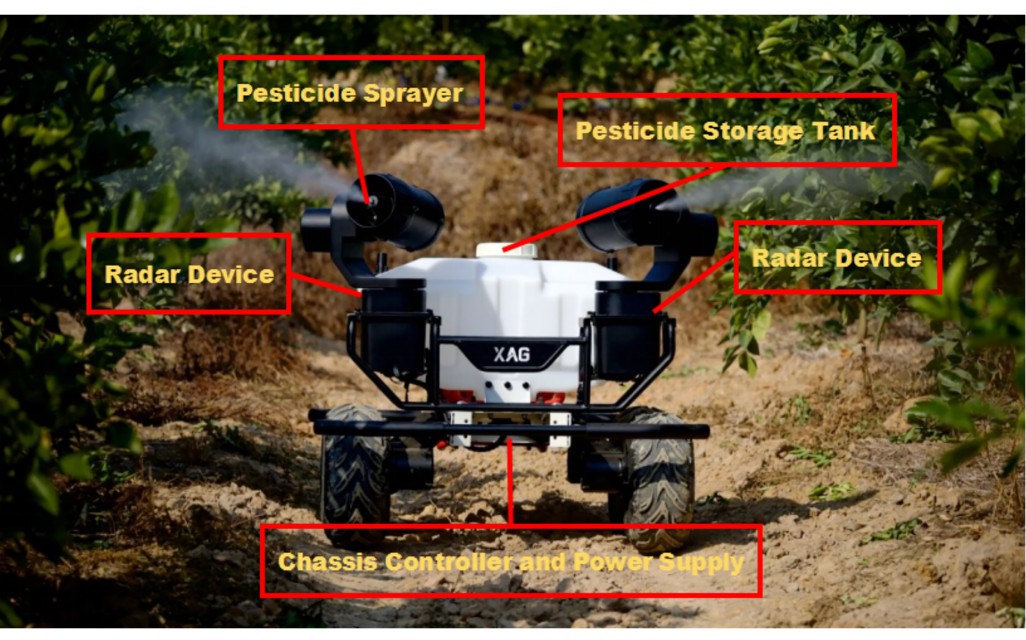

**Figure 8 Structure of wheeled plant protection robot.**

**Table 2 List of benchmark function parameters.**

| Benchmark functions | Dimension | Range | Theoretical minimum value |
|---|---|---|---|
| $f_1(x) = \sum_{i=1}^{n} x_i^2$ | 30 | $[-100, 100]$ | 0 |
| $f_5(x) = \sum_{i=1}^{n-1}\left[100\left(x_{i+1} - x_i^2\right)^2 + (x_i - 1)^2\right]$ | 30 | $[-30, 30]$ | 0 |
| $f_8(x) = \sum_{i=1}^{n} -x_i sin(\sqrt{|x_i|})$ | 30 | $[-500, 500]$ | 0 |
| $f_{13}(x) = 0.1\{sin^2(3\pi x_1) + \sum_{i=1}^{n}(x_i - 1)^2[1 + sin^2(3\pi x_i + 1) + (x_n - 1)^2[1 + sin^2(2\pi x_n)]\} + \sum_{i=1}^{n} u(x_i, 5, 100, 4).$ | 30 | $[-50, 50]$ | 0 |
| $f_{15}(x) = \sum_{i=1}^{11}\left[a_i - \frac{x_1\left(b_i^2 + b_1 x_2\right)}{b_i^2 + b_1 x_3 + x_4}\right]^2$ | 4 | $[-5, 5]$ | 0.1484 |
| $f_{17}(x) = \left(x_2 - \frac{5.1}{4\pi^2}x_1^2 + \frac{5}{\pi}x_1 - 6\right)^2 + 10\left(1 - \frac{1}{8\pi}\right)cos x_i + 10$ | 2 | $[-5, 5]$ | 0.3 |

Optimization Algorithm based on probability selection (MWOA) (*Wei, Li & Zhang, 2023*), the algorithm that combines the Harmony Search algorithm with the Whale Optimization Algorithm (HSWOA) (*Li et al., 2020*), the cosine adapted modified whale optimization algorithm (CamWOA) to reduce iteration step sizes (*Saha et al., 2022*), and the Whale Optimization Algorithm with adaptive weights (WWOA) (*Cheng, Wang & Wang, 2022*). Additionally, 20 comparative experiments were conducted using six typical benchmark test functions to objectively assess the robustness and effectiveness of the algorithm improvement based on the average convergence curve of the fitness function. Secondly, in a simulation environment, six environmental maps with varying starting points, target points, and obstacles' numbers and locations were selected as testing scenarios to assess the

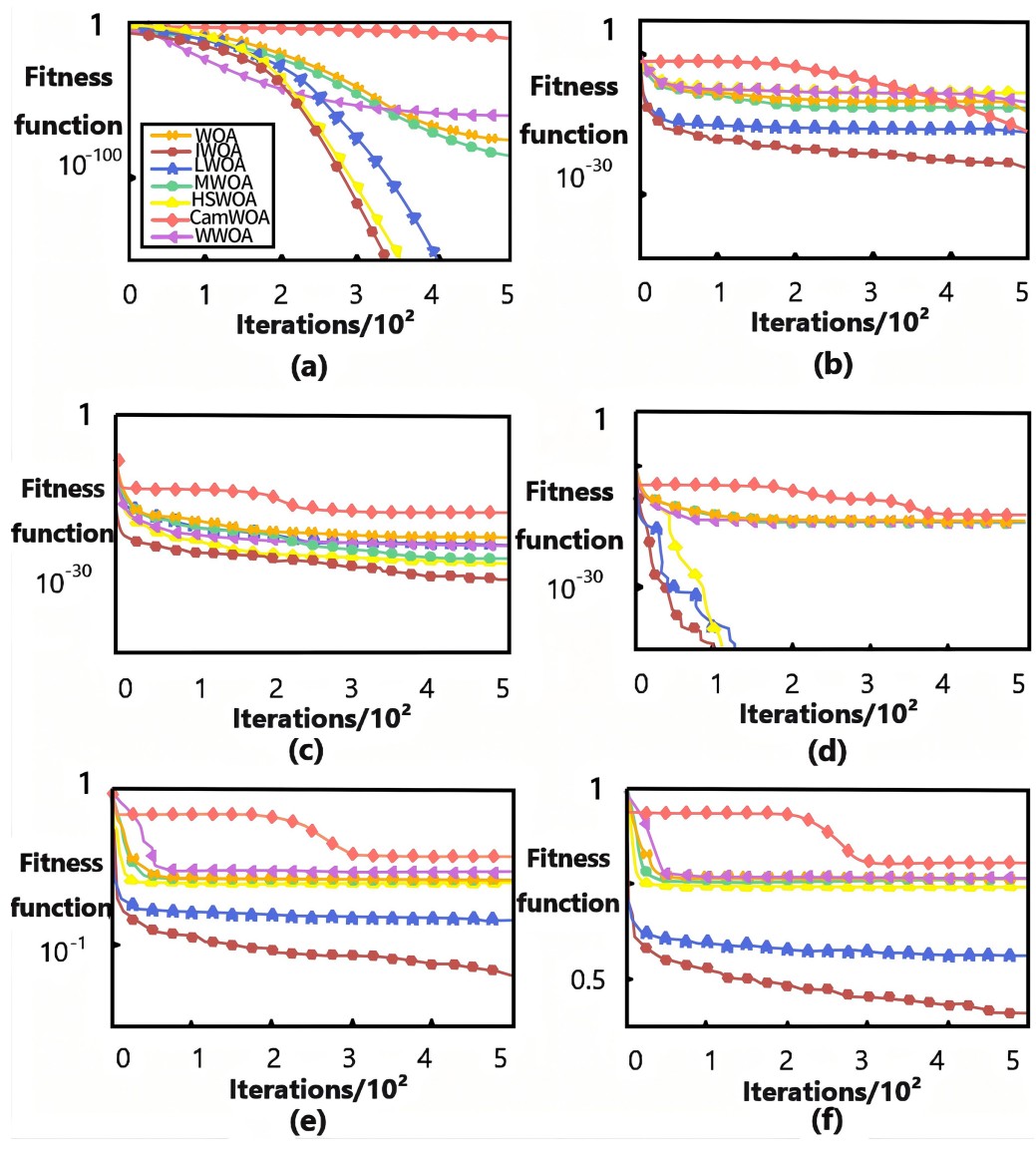

**Figure 9 Average convergence curve of fitness of test function.** (A) $f_1$ curves; (B) $f_5$ curves; (C) $f_8$ curves; (D) $f_{13}$ curves; (E) $f_{15}$ curves; (F) $f_{17}$ curves.

path improvement of the A*-IWOA algorithm (*Wang et al., 2022*). The algorithm's effectiveness was verified by comparing its performance with the traditional A* and RRT algorithms in terms of running time, path length, number of turning points, and energy consumption. Finally, to validate the effectiveness of the A*, RRT, standard A*-IWOA, and improved A*-IWOA algorithms, a physical experiment was conducted in a mountainous orchard in Gansu.

## Performance testing of the improved IWOA algorithm

In this experiment, six commonly used benchmark test functions from the IEEE CEC benchmark test set were used, covering unimodal, multimodal, and composite functions, as presented in Table 2. The population size was set to 30, with a maximum number of

**Table 3 Test results data.**

| Function | Algorithms | Optimal | Worst | Average | Standard | Consuming/s |
|---|---|---|---|---|---|---|
| $f_1$ | Improved IWOA | 0.00e+00 | 0.00e+00 | 0.00e+00 | 0.00e+00 | 1.0233 |
| | WOA | 1.47e−178 | 3.01e−161 | 2.35e−173 | 4.38e−170 | 1.9038 |
| | LWOA | 0.89e−121 | 2.75e−149 | 1.78e−151 | 8.41e−201 | 1.7544 |
| | MWOA | 1.02e−130 | 7.88e−171 | 6.07e−154 | 1.34e−149 | 1.7068 |
| | HSWOA | 1.02e−131 | 2.00e−161 | 2.32e−158 | 5.18e−180 | 1.2331 |
| | CamWOA | 6.67e−118 | 3.01e−152 | 8.35e−133 | 3.47e−170 | 1.1138 |
| | WWOA | 3.47e−150 | 5.71e-131 | 2.39e−143 | 7.33e−172 | 1.1189 |
| $f_{13}$ | Improved IWOA | 0.00e+00 | 0.00e+00 | 0.00e+00 | 0.00e+00 | 0.8724 |
| | WOA | 1.32e−201 | 3.81e−191 | 7.55e−193 | 3.18e−200 | 1.2008 |
| | LWOA | 0.78e−181 | 2.21e−169 | 1.18e−171 | 2.40e−206 | 1.0044 |
| | MWOA | 1.41e−200 | 7.01e−181 | 9.11e−194 | 1.34e−221 | 1.0068 |
| | HSWOA | 3.01e−191 | 7.36e−164 | 2.31e−178 | 5.10e−200 | 1.2071 |
| | CamWOA | 8.67e−218 | 3.71e−152 | 8.31e−183 | 3.47e−190 | 1.1130 |
| | WWOA | 3.41e−190 | 6.93e−211 | 2.32e−203 | 7.33e−221 | 0.9189 |
| $f_{15}$ | Improved IWOA | 0.20e+00 | 0.09e+00 | 0.10e+00 | 0.03e−02 | 0.9331 |
| | WOA | 1.32e+00 | 7.12e+00 | 1.39e+00 | 1.18e−110 | 1.7328 |
| | LWOA | 0.73e+00 | 0.99e+00 | 0.80e+00 | 2.40e−106 | 1.1114 |
| | MWOA | 0.71e+00 | 1.21e+00 | 1.01e+00 | 1.14e−121 | 1.2260 |
| | HSWOA | 0.43e+00 | 0.93e+00 | 0.99e+00 | 2.91e−100 | 1.2099 |
| | CamWOA | 0.67e+00 | 1.91e+00 | 1.45e+00 | 1.47e−090 | 1.9130 |
| | WWOA | 0.49e+00 | 0.88e+00 | 0.78e+00 | 5.49e−121 | 1.1181 |

iterations of 500. Since algorithm dimensionality significantly affects optimization performance, the dimensions of the six test functions in Table 2 vary from 2 to 30, providing a comprehensive test of the algorithm's solving ability across low to high dimensions.

Figure 9 shows the average convergence curves of the fitness functions, obtained by running each of the six benchmark test functions 20 times. The $f_1$ and $f_5$ curves evaluate the algorithm's development ability (Figs. 9A, 9B), $f_8$ and $f_{13}$ assess its search ability (Figs. 9C, 9D), and $f_{15}$ and $f_{17}$ reflect its overall optimization ability (Figs. 9E, 9F). Due to the introduction of Circle mapping for uniformly distributed population positions and inertia dynamic adjustment of weights, the algorithm better avoids local optima. The improved IWOA algorithm demonstrates superior convergence speed and accuracy in solving unimodal, multimodal, and composite functions compared to other algorithms, confirming the effectiveness of the improved algorithm.

Table 3 provides performance data summarizing the results of the test functions described above, where the optimal, worst, and average values reflect the algorithm's optimization ability and effectiveness, while the standard deviation indicates its stability. As shown in Table 3, the improved IWOA achieved theoretical optimal values of 0 with the shortest time when solving the unimodal function $f_1$. For multimodal function $f_{13}$, the

**Table 4  The kinematic parameters of the robot.**

| Maximum limit | Value |
|---|---|
| Maximum linear speed (m/s) | 1.5 |
| Maximum angular velocity (rad/s) | 0.8 |
| Maximum angular acceleration (rad/s$^2$) | 0.3 |
| Maximum linear acceleration (m/s$^2$) | 0.4 |

**Table 5  The initial parameter settings of improved A*-IWOA.**

| Initial parameters | Value |
|---|---|
| Grid environment | 20 × 20/30 × 30 |
| Starting point coordinates | 35 |
| End point coordinates | 285 |
| Initial population size | 30 |
| Maximum number of iterations | 500 |

improved IWOA significantly accelerated convergence by 20% compared to other WOA variants. For composite function $f_{15}$, the improved IWOA randomly could calculate the changes in dimensions, and its multiple indicators approached a theoretical value of 0.1484, with a slight increase in the time consumption, yet still the fastest among the different algorithms. This indicates that the optimization performance and efficiency of the improved IWOA are significantly enhanced by dynamically adjusting the position and inertia weight of the uniformly distributed population using Circle mapping.

## DISCUSSION

### Path planning testing in 2.5D elevation grid map simulation

This simulation experiment was conducted on a Windows 10 system with 32 GB of memory, a 2.9 GHz CPU, and a Matlab R2021b programming workstation. The robot's kinematic parameters and the initial settings of the improved A*-IWOA algorithm are presented in Tables 4 and 5. In this investigation, three grid maps were tested of sizes 20 × 20, 30 × 30, and 50 × 50, where each cell array represents the horizontal, vertical, and elevation values at its center point. In the simulation process, the cell elevation values were represented by the hue H value in the HSV model, where a higher hue H value indicates a greater elevation. For example, in Fig. 3, a purple cell has a higher elevation than a red cell. Following the path search logic described in this article, in the main search area of the path nodes, cell elevation values were distributed randomly for the simulation test (*Akay & Karaboga, 2010*; *Al-Dabbagh et al., 2014*). Four scenarios of plant protection robots were constructed, and the A*, RRT, A*-IWOA, and improved A*-IWOA algorithms were applied for path planning simulations, with results shown in Fig. 10.

Figures 10 (1-1)–(1-4) highlight that the occupancy rate of obstacles in the robot passage area is 20% with a grid map of 20 × 20. In Fig. 10 (2-1) to Fig. 10 (2-4), the grid

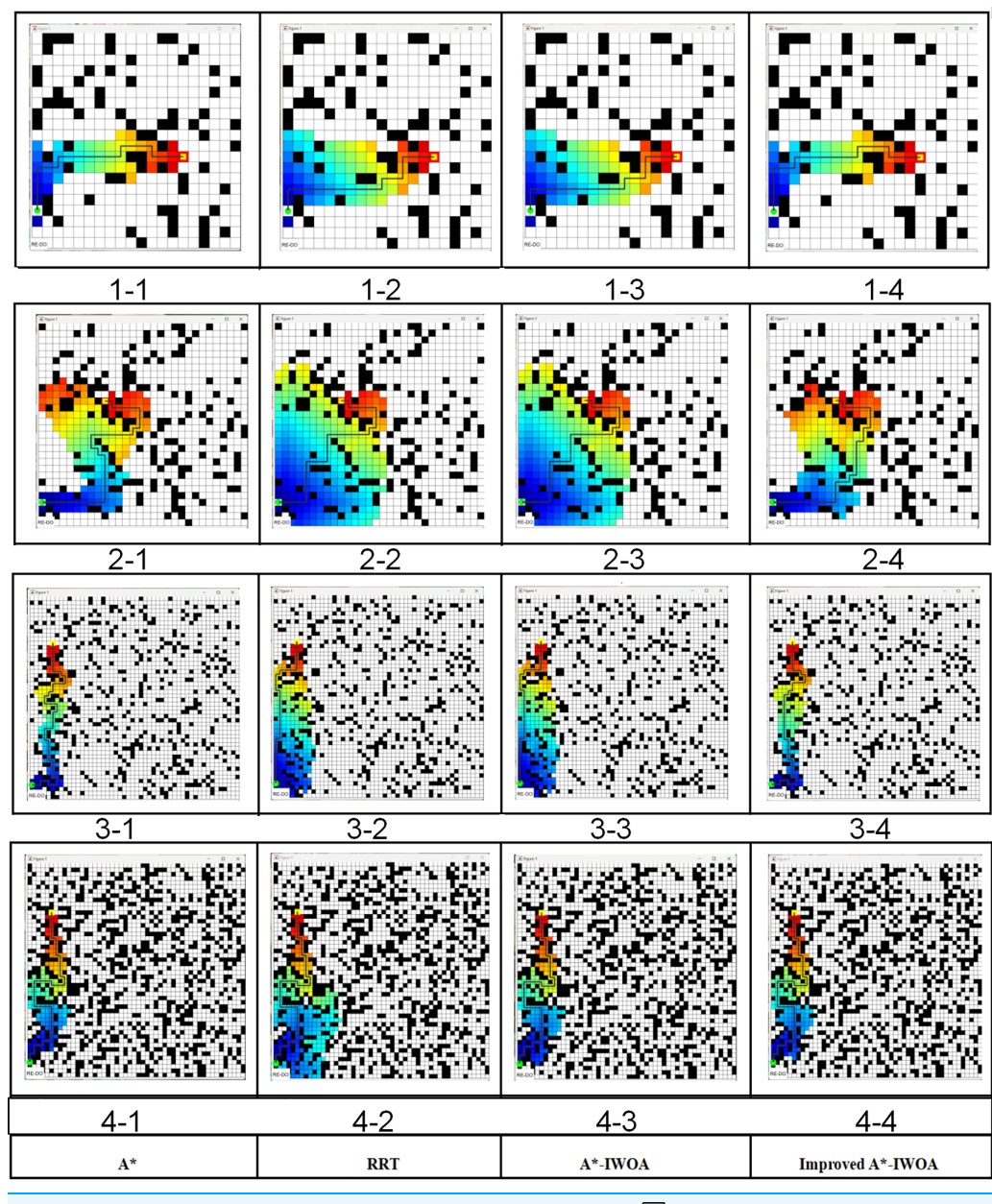

**Figure 10 The path planning test results.**

map is 30 × 30 with a 20% obstacle occupancy rate. In Fig. 10 (3-1) to Fig. 10 (3-4), the grid map is 50 × 50 with a 29% obstacle occupancy rate, and in Fig. 10 (4-1) to Fig. 10 (4-4), the grid map is 50 × 50 with a 40% obstacle occupancy rate. The solid black lines represent the planned paths of each algorithm. Differences in path length, number of turning points, and calculation time among the four methods are presented in Table 6. It is noted that the improved A*-IWOA algorithm showed a notable reduction in path length, averaging 5.7%, 9.2%, and 5.1% shorter than the A*, RRT, and A*-IWOA algorithms, respectively. In terms of turning points, the improved A*-IWOA algorithm performed equally well as RRT in larger grids but outperformed other algorithms in smaller grids. In calculation time, the

**Table 6 Road planning result data for four scenes.**

| Scenes | Algorithms | Planning length/m | Turning points | Calculation time/ms |
|--------|-----------|-------------------|----------------|---------------------|
| 1 | A* | 33.0 | 8 | 1,300 |
| | RRT | 34.0 | 5 | 2,247 |
| | A*-IWOA | 33.4 | 9 | 1,210 |
| | Improved A*-IWOA | 27.0 | 5 | 790 |
| 2 | A* | 30.1 | 11 | 1,272 |
| | RRT | 30.0 | 10 | 2,368 |
| | A*-IWOA | 30.3 | 11 | 1,205 |
| | Improved A*-IWOA | 29.0 | 8 | 881 |
| 3 | A* | 37.4 | 13 | 1,283 |
| | RRT | 39.2 | 8 | 1,762 |
| | A*-IWOA | 34.2 | 10 | 1,270 |
| | Improved A*-IWOA | 30.0 | 6 | 817 |
| 4 | A* | 33.4 | 19 | 1,290 |
| | RRT | 38.2 | 18 | 2,982 |
| | A*-IWOA | 30.0 | 17 | 1,209 |
| | Improved A*-IWOA | 27.0 | 13 | 964 |

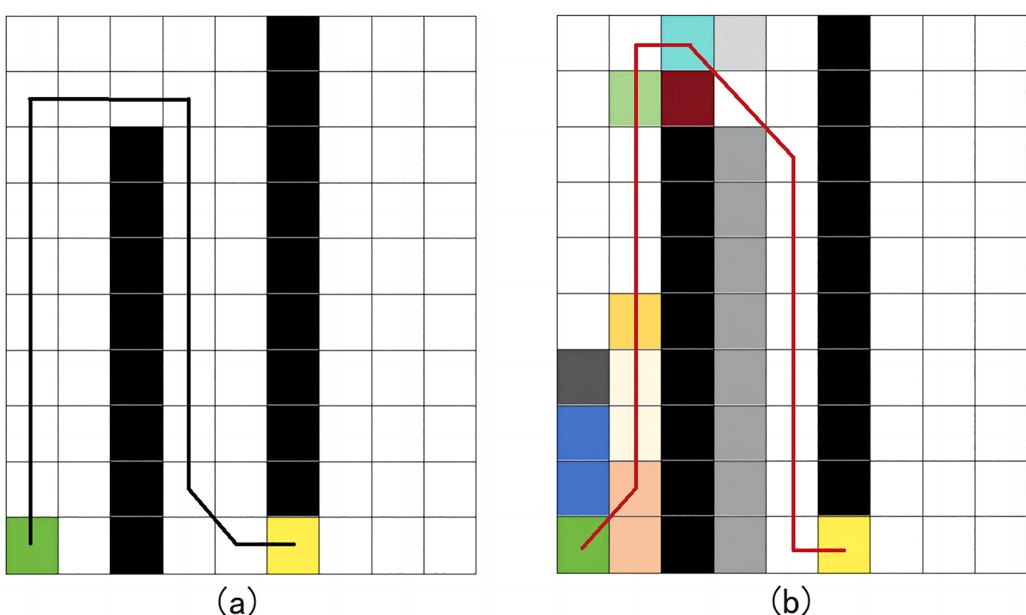

(a)                          (b)

**Figure 11 Comparison of planning effects on 2D and 2.5D maps.** (A) 2D map planned path; (B) 2.5D map planned path.

improved A*-IWOA algorithm was 12.2%, 20.2%, and 15.6% faster than A*, RRT, and A*-IWOA, respectively, on average.

To provide a realistic representation of the inter-row working environment for fruit trees in mountainous environments, a 10 × 10 2.5D elevation grid map was used. Black

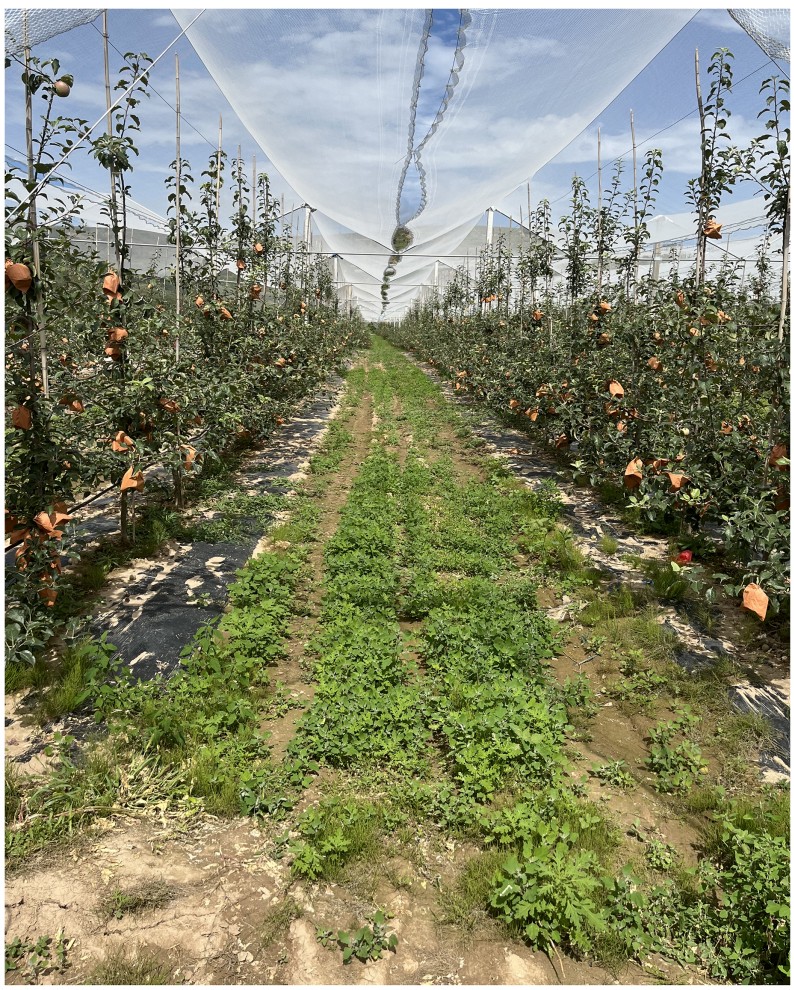

**Figure 12  Real environment.**               

obstacles represent the inter-row positions of fruit trees, and different grid colors indicate the vertical heights at their positions. The green grid marks the robot's starting position, while the yellow grid represents the target position. Using Matlab R2021b programming workstation, the path planning effect of the improved A\*-IWOA algorithm was tested on both 2D and 2.5D maps. Experimental results are shown in Fig. 11, highlighting notable differences in the planned paths. In Fig. 11B, due to the varying road surface vertical heights, the robot tends to choose and move forward with paths along lower terrain.

## Experiment with robot operation path planning in a real orchard

To verify the effectiveness of the improved algorithm, a physical experiment was conducted in a mountainous orchard in Gansu, as shown in Fig. 12. In this orchard, the plant spacing is 20–30 cm, and the row spacing is 70–80 cm. The altitude ranges from 500 to 1,000 m, with a relative height of no more than 200 m, and the slope is relatively gentle. This allows for adjustments to the camera and radar-ranging height of the plant protection robot, ensuring that there are at least five plants within the camera's field of view at the

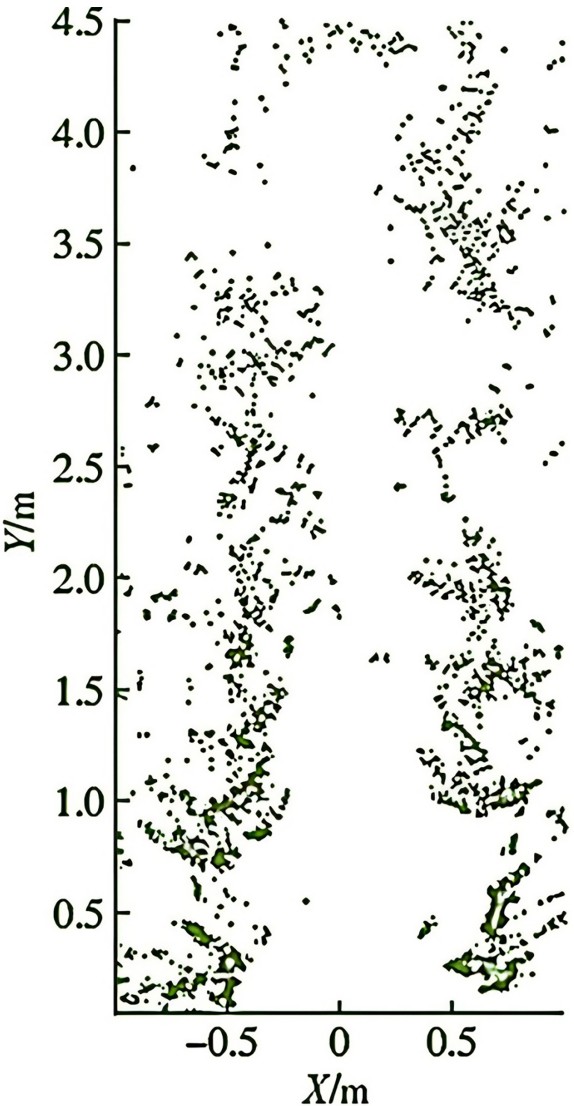

**Figure 13  Real orchard point clouds.**     

robot's speed limit. During the fruit ripening period, the orchard's visual environment consists of green trees, red fruits, gray-brown ground, and a gray-blue sky, providing clear recognition of the machine's mission. The K-means clustering method was used to identify the position of the main trunk of the fruit tree (as shown in Fig. 13). Additionally, the navigation line was planned by defining the communicable area through the central area. Variations in width between 70 and 80 cm had minimal impact on recognition error within the central area. In this experiment, the maximum navigation line error was 7.07 cm, the minimum error was 0.5 cm, and the average error was 3.1 cm.

Furthermore, in the physical environment, the A*, RRT, standard A*-IWOA, and improved A*-IWOA algorithms were each loaded onto the robot ROS2 platform, and the path planning effects are shown in Fig. 14. In the figure, the cluster centers of fruit trees were extracted using the K-means clustering algorithm from the LiDAR point cloud data, and the row size of fruit trees was determined based on the least squares method, indicated

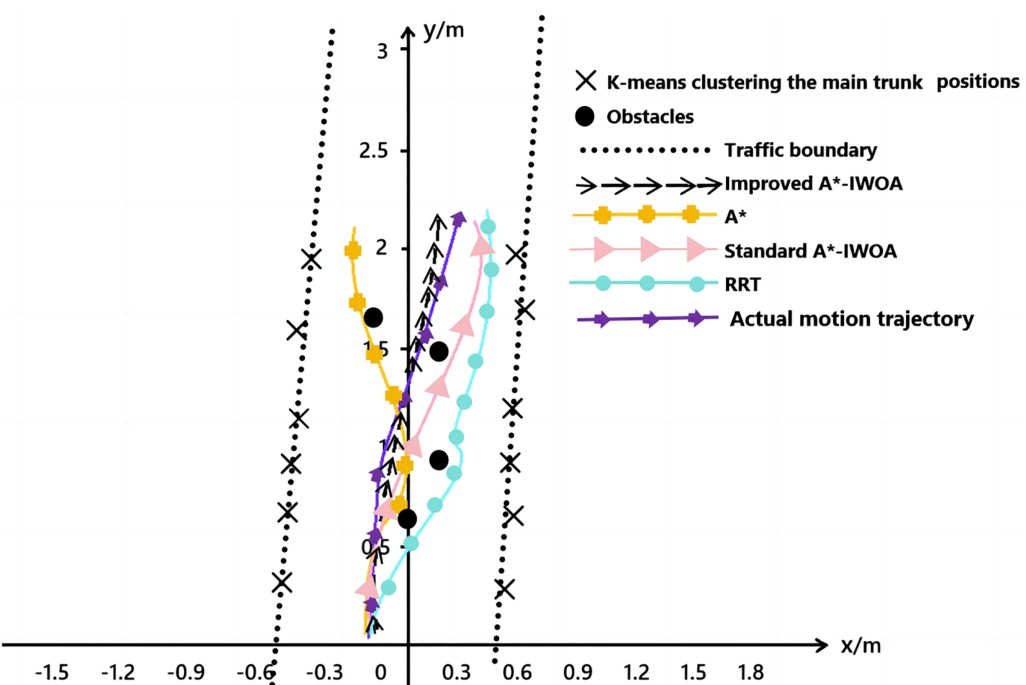

**Figure 14 Comparison of four algorithms for path planning in real orchard.**

by the black dotted line in the figure. The obstacle positions, based on point cloud data, are marked by black dots in the figure. It was observed that the improved A\*-IWOA algorithm selected a more direct, non-detour path, while the other three algorithms chose a longer, more circuitous route around obstacles. By matching the spatial point cloud position information, the robot's trajectory information, derived from the improved A\*-IWOA algorithm, is overlaid on the image and shown by the purple line. This trajectory closely resembles the path obtained by controlling the robot between rows of fruit trees *via* the upper computer, demonstrating the high feasibility of the improved A\*-IWOA algorithm.

## CONCLUSION

For the unstructured work environment of plant protection robots in mountainous orchards, this article proposes an 8-domain A\* path search algorithm that incorporates a vector cross-product decision value based on the robot's energy consumption model within a 2.5D elevation grid map. The dynamic weight factor is optimized using the IWOA algorithm through dynamic adjustment of uniform population position and inertia weight. Compared to a 2D grid map environment, the path planning speed is significantly improved, particularly with the introduction of energy consumption models and optimization of the cost function using vector cross-product decision value factors. The resulting path achieves bidirectional optimization in terms of energy consumption and path length, greatly enhancing the path planning effect and computational efficiency.

The article presents performance testing and path planning experiments of the improved algorithm conducted in both the Matlab R2021b simulation environment and an

actual orchard operation scenario using the ROS 2 system. The robustness and effectiveness of the proposed algorithm are compared with those of WOA, LWOA, MWOA, HSWOA, and others by analyzing the average convergence curve of the fitness function under six typical benchmark test functions. Additionally, path planning simulations were conducted for A*, RRT, standard A*-IWOA, and improved A*-IWOA across six representative scenarios, with a focus on path planning differences between 2D and 2.5D grid maps. Finally, a physical experiment was performed in a mountainous orchard in Gansu province to validate the effectiveness of the improved algorithm. Results demonstrate that the improved algorithm offers significant advantages in computational accuracy, convergence speed, and efficiency. Moreover, the planned path meets energy consumption and path planning requirements for working robots in unstructured mountain environments. While the article discusses the advantages of the improved algorithm in detail, factors such as variations in orchard environments, standardization of planting, and the computational power of the robot's motherboard could affect the path planning outcome. The improved algorithm for path planning in plant protection robots proposed in this article could be replicated and promoted in other fields, such as picking robots, factory inspection robots, and more complex environments.

Based on the findings attained in this study, future research will focus on the following areas:

(1) Further improving the algorithm to enhance path planning effectiveness;

(2) Developing methods to enable detection and navigation of plant protection robot operation channels under various environmental interferences;

(3) Implementing autonomous navigation operations for plant protection robots inspired by neuroscience principles.

### Funding

This work was supported by the Innovation Fund for College Teachers in Gansu Province No. 2013A-114, the Tianshui Normal University Industry Support and Guidance Project No. CYZ2023-05, and the Tianshui Normal University Innovation and Entrepreneurship Project No. CXCYJG-JGXM202304JD. The funders had no role in study design, data collection and analysis, decision to publish, or preparation of the manuscript.

### Grant Disclosures

The following grant information was disclosed by the authors:
Innovation Fund for College Teachers in Gansu Province: 2013A-114.
Tianshui Normal University Industry: CYZ2023-05.
Tianshui Normal University Innovation and Entrepreneurship: CXCYJG-JGXM202304JD.

### Competing Interests

The authors declare that they have no competing interests.

## Author Contributions

- Jing Niu conceived and designed the experiments, performed the computation work, authored or reviewed drafts of the article, and approved the final draft.
- Chuanyan Shen conceived and designed the experiments, performed the experiments, prepared figures and/or tables, and approved the final draft.
- Lipeng Zhang analyzed the data, authored or reviewed drafts of the article, and approved the final draft.
- Qijun Li performed the experiments, prepared figures and/or tables, and approved the final draft.
- Haohao Ma analyzed the data, performed the computation work, authored or reviewed drafts of the article, and approved the final draft.

## Data Availability

The data is available at Zenodo: Jing, N. (2024). Python A star-IWOA_1.7z [Data set]. Zenodo. https://doi.org/10.5281/zenodo.14264017.

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
