# Peer review of "A multi-objective path optimization method for plant protection robots based on improved A*-IWOA"

_PeerJ Computer Science, doi:10.7717/peerj-cs.2620_

## Round 0.1 · original submission · Major Revisions

Dear authors,

Thank you for submitting your manuscript for review. According to the 9 reviewers' comments, the manuscript requires Major Revisions. Please read carefully at the reviewers' comments and make revisions correspondingly. In particular, most figures of you manuscript are too blurry and copyrighted elements should be replaced with your own if there is any.

All the best.

·

Basic reporting

1. The introduction broadly describes the problem of energy consumption in plant protection robots operating in mountainous environments but lacks a specific contextual framework. It is essential to provide more detailed background on the specific challenges faced in such environments. Additionally, clarifying why the proposed solution is superior to existing methods would enhance the introduction's impact.

2. While the introduction identifies the need for path optimization, it does not sufficiently justify the choice of the improved A*-IWOA method over other algorithms. The authors should clearly state why the A*-IWOA combination was chosen and how it uniquely addresses the challenges identified, distinguishing it from other potential approaches.

3. Terms such as "2.5D elevation grid map" and "cross product decision values" are introduced without adequate explanation. Readers unfamiliar with these specific concepts may find it challenging to understand the significance of the proposed method. Including concise definitions or explanations for these terms would improve clarity.

4. The research objectives are not clearly delineated. The introduction should explicitly state the research goals, such as improving path planning efficiency, reducing energy consumption, or enhancing operational accuracy. Clearly defined objectives would help readers understand the study's purpose and anticipated contributions.

5. The introduction delves too quickly into the technical specifics of the proposed method (e.g., IWOA and A* algorithms), which may overwhelm readers unfamiliar with these techniques. A more balanced approach, first outlining the broader problem and then gradually introducing technical specifics, would be more effective.

6. The related literature section references several works on path optimization and energy consumption but lacks a comprehensive review of recent studies that directly relate to the proposed method. A broader survey of current research, particularly those involving hybrid algorithms like A*-IWOA, would provide a stronger foundation for the study.

7. Connection between the reviewed literature and the current study's objectives is not clearly established. Each cited work should be more explicitly tied to the specific aspects of the proposed method, showing how they contribute to the development of the research question or methodology.

8. Given that energy consumption is a critical component of the study, the literature review should more thoroughly explore existing models and methods used to evaluate and optimize energy consumption in autonomous robots. This would help position the current study more effectively within the existing body of knowledge.

Experimental design

1. Methodology lacks comprehensive benchmarking against state-of-the-art algorithms beyond a few variations of the A* and WOA algorithms. The authors should consider comparing the improved A*-IWOA against other path planning and optimization algorithms, such as RRT* or Dijkstra, to provide a broader perspective on its performance .

2. Yes, the article briefly mentions tests conducted in a "mountainous orchard scene," but the adaptation of the algorithm to this real-world scenario is inadequately detailed. A more thorough discussion is needed on how environmental variables (e.g., terrain roughness, plant distribution) were accounted for in the algorithm's design and testing.

3. Heavy reliance on simulation results to validate the improved A*-IWOA algorithm without adequately cross-referencing these findings with real-world data. Additional validation through field experiments or comparison with known benchmarks is needed to support the claims made about the algorithm's performance.

4. The paper highlights minor improvements in certain performance metrics, such as a slight reduction in path length or a marginal increase in convergence speed. The results should focus on the most significant findings and contextualize their impact on the broader field of autonomous robotics.

5. The interpretation of some results, such as those concerning the "optimal path length" or "energy efficiency," is vague and lacks depth. More detailed interpretation and discussion of what these results mean for the field of autonomous robotics would enhance the paper's contribution.

Validity of the findings

1. Please address your algorithms' limitations or cases where it underperforms. A balanced discussion should include potential weaknesses, such as scenarios where the algorithm might not be effective, to provide a more nuanced understanding of its applicability​.

2. Discussion does not adequately address alternative methods or algorithms that could also be suitable for the problem at hand. The authors should consider comparing their approach to other recent advancements in path optimization, such as machine learning-based or hybrid optimization methods​.

3. The conclusions state that the algorithm "significantly improves" path planning efficiency and computational effectiveness, but these claims are not quantified or contextualized. The authors should provide specific data points or comparisons to substantiate these claims​.

4. The conclusions fail to address whether the proposed algorithm can be scaled or generalized to other applications or environments. The authors should discuss the potential for wider application and any necessary modifications to achieve it​.

Additional comments

There are many questions about this research that needs to be answered first to become worthy for publication. I am heavily recommending major revisions based on my experience. Thank you very much.

·

Basic reporting

In the introduction section the authors accurately describe the algorithms used by robots to find the optimal path. The article provides relevant literature references to these algorithms. However, I would like to advise the authors to slightly expand the introduction by describing thoroughly the area of application (agricultural robotics) and the features of agricultural robots in complex terrain.
The structure of the article differs from the PeerJ Standard Sections. There are not Materials & Methods, Results, and Discussion sections. However, this is not a critical objection. The work describes in detail the robot model, the developed algorithm, and the analysis of its effectiveness. This is enough to understand the goals and results of the study conducted by the authors.
The work provides many figures, which significantly simplify the understanding of the article. However, all the figures have a common problem - low quality of the figures. This may be due to the use of image compression while preparing the article. It is necessary to replace the figures with higher quality (higher resolution and with a lower compression ratio). Moreover, there are some minor comments on the design of figures and tables (are given in General comments).
The authors of the paper presented the source code with the implementation of the A*-IWOA algorithm in the Python and Matlab programming languages. The presented files contain the implementation of a number of algorithms discussed in the article, tests for assessing their effectiveness and the ability to display animation on the screen. Some of the code has comments. As a disadvantage of the source files, we can note the lack of translation of the README.md file. Translation into English would significantly facilitate the work with the source code.

Experimental design

The topic of the study is relevant and corresponds to the Aims & Scope of the PeerJ Computer Science journal.

Validity of the findings

The authors tested the developed algorithm in the Matlab environment. The presented materials are sufficient to illustrate the experiment. It is also worth noting the presence of an experiment in real conditions (in a mountainous orchard scene in Gansu province, section 5.4). It bears emphasis that the experiment in real conditions is extremely interesting for researchers and perhaps it was worth describing it in more detail. In particular, give a detailed description of the robot structure (model of engines, battery, etc.). Add information about the relief of the study area (for example, in the form of a 2.5D elevation grid map) and a photograph of the garden from a higher angle. And also provide the results of measuring energy costs for different trajectories, if such measurements were carried out. This comment is advisory.

Additional comments

The article «A Multi-Objective Path Optimization Method for Plant Protection Robots based on Improved A*-IWOA» can be published after making minor edits:
1. Part of the formula for the benchmark function f13(x) is not visible in Table 2, it should be added.
2. In Table 4, you need to add units of measurement for the specified robot parameters (km/h, etc.).
3. In the table with pseudocode (Table 1), the line numbers should be put in a separate column, then the code will be even more clear.
4. In Figure 7, FALSE should probably be used instead of FLAUSE. In addition, the "When a whale ..." block should be slightly enlarged - so that the inscription fits completely.
5. In the graphs with the fitness function (Figure 9), it is worth using colored lines, which will improve the perception of the graphs. In a black and white image, the lines for different functions merge.
6. In Figure 10, each scene shows trajectories described in the text (lines 408-411). This description should be duplicated in the figure caption.
7. In Figure 14, units of measurement should be indicated (apparently, these are meters). Also, for clarity, it is worth highlighting the trajectories in different colors.
8. In equations (11), (12), the name of the tan function merges with the name of the argument, which interferes with perception. Perhaps it is worth changing the spelling of the function (remove the italics) or use highlighting with spaces or brackets (tan(α)).
9. Equations (22) and (23) are centered, unlike the other formulas, they should be aligned to the left edge.

In addition, the authors are recommended, if possible:
1. In the introduction, consider in more detail the specifics of agricultural robotics and the use of robots in mountainous conditions.
2. Describe in more detail the experiment conducted in the mountain garden (provide more photographs and an elevation map of the experimental site).
3. Improve the quality of the figures.
4. In the source code, translate the README.md file into English.

·

Basic reporting

No comment

Experimental design

No comment

Validity of the findings

No comment

Additional comments

The overall structure of the paper is satisfactory; however, it addresses only a small portion of the spraying robot. I would have preferred to see more detailed insights into how the robot integrates image acquisition with the spraying process. Nonetheless, the current version is acceptable. I have provided some minor suggestions, which can be found in the attached PDF. Best regards.

---

## Round 0.2 · Minor Revisions

Please address the final issues from the reviewer, relating to readability and flow

·

Basic reporting

1. The paper needs to improve the clarity and flow of sentences. Just a minor revision.

Experimental design

1. Queries are answered reasonably and responsibly.

Validity of the findings

1. Findings are now improved and easy to understand.

Additional comments

1. Please improve the cohesiveness for each sentences and paragraphs.

·

Basic reporting

no comment

Experimental design

no comment

Validity of the findings

no comment

Additional comments

The authors have corrected all the identified shortcomings. The article corresponds to requirements of the PeerJ Computer Science.

·

Basic reporting

It is accepted.

Experimental design

It is accepted.

Validity of the findings

It is accepted.

Additional comments

It is accepted.

---

## Round 0.3 · accepted · Accept

The article has been updated based on the feedback provided to align with the reviewers' comments and expectations.

·

Basic reporting

1. Manuscript is all good now.

Experimental design

1. No more comments.

Validity of the findings

1. Substantially improved already.

Additional comments

1. Ready for publication.